# Dentin Bonding of TheraCal LC Calcium Silicate Containing an Acidic Monomer: An In Vitro Study

**DOI:** 10.3390/ma13020293

**Published:** 2020-01-08

**Authors:** Young Kyung Kim, Mi-Hee Hong, Tae-Yub Kwon

**Affiliations:** 1Department of Conservative Dentistry, School of Dentistry, Kyungpook National University, Daegu 41940, Korea; wisekim@knu.ac.kr; 2Department of Orthodontics, School of Dentistry, Kyungpook National University, Daegu 41940, Korea; mhhong1208@knu.ac.kr; 3Department of Dental Biomaterials, School of Dentistry and Institute for Biomaterials Research & Development, Kyungpook National University, Daegu 41940, Korea

**Keywords:** calcium silicate, pulp capping, dentin bonding, acidic monomer

## Abstract

The purpose of this study was to evaluate whether the incorporation of an acidic monomer into the pulp-capping material TheraCal LC, which has a weak dentin bonding, increases the shear bond strength (SBS) to dentin. Di-2-hydroxyethyl methacryl hydrogenphosphate was incorporated into the material at 0.0 (TL0, control), 5.0 (TL5), and 10.0 (TL10) wt%. The water contact angle (CA) and mechanical properties for each material were also studied (*n* = 6). Debonding was performed at two different times (immediate and after 24 h) (*n* = 12). Hydroxyl and calcium ion releases in water at 37 °C were monitored up to 28 days (*n* = 6). The addition of the acidic monomer decreased the CAs (*p* < 0.001) and increased the flexural moduli (*p* < 0.001). The debonding time did not significantly affect the SBS values (*p* = 0.600). The TL10 group exhibited the highest SBS values, followed by the TL5 group. The TL10 group released significantly more calcium ions than the other two groups from 3 days (*p* < 0.05). The incorporation of the acidic monomer at 10.0 wt% into TheraCal LC enhanced bonding to dentin, while not negatively affecting the mechanical properties and ion-leaching capacity of the material.

## 1. Introduction

Direct pulp capping involves covering the exposed pulp with a material to preserve its vitality and its functional and biological activities [1]. Calcium hydroxide has been used in direct/indirect pulp-capping mainly due to its ability to release hydroxyl and calcium ions on dissolution [1,2]. However, its high solubility, lack of adhesion, and poor physical–chemical properties have recently gained relevance as clinical issues [2].

Consequently, mineral trioxide aggregate (MTA) and MTA-derived materials have been suggested as new direct/indirect pulp-capping agents and have shown favorable results in clinical trials [2]. It has been reported that MTA is more effective than calcium hydroxide due to its enhanced interaction with pulp tissue and fewer negative responses to pulp [1]. However, MTA’s long setting time remains one of its main drawbacks, despite many studies aiming to address this clinical disadvantage [1,2].

TheraCal LC (Bisco, Schamburg, IL, USA) is a resin-modified calcium silicate base/liner used for pulp capping [1]. Its visible light-cured set may permit a dentist immediate placement and the condensation of a restorative material. Gandolfi et al. [1] reported that the material showed a lower solubility and a higher calcium ion release than ProRoot MTA (Dentsply, Johnson City, TN, USA) and Dycal (Dentsply). However, TheraCal LC has low bond strength to dentin [3], and the stability may be impaired when the restorative material is placed and condensed over it [4]. Its dentin bonding could be significantly improved by the use of dentin adhesives or resin cements [3,4]. Nonetheless, it would be more clinically convenient to utilize a calcium silicate material with self-adhesive function.

In this in vitro study, we investigated the effect of incorporation of a functional monomer into TheraCal LC on the shear bond strength (SBS) to dentin. The mechanical properties of the modified materials were compared with the original material. pH and calcium release from the original and modified materials in water were also investigated over 28 days. The first null hypothesis was that the addition of a functional monomer into TheraCal LC would not enhance the SBS. The second null hypothesis adopted was that the functional monomer addition would not affect the mechanical properties nor the ion release tendency of the material.

## 2. Materials and Methods

### 2.1. Material Preparation

Two modified TheraCal LC materials (groups TL5 and TL10) were prepared from the original product by incorporating 5.0 and 10.0 wt% di-2-hydroxyethyl methacryl hydrogenphosphate (di-HEMA phosphate) from Sigma-Aldrich (St. Louis, MO, USA), respectively. The original TheraCal LC (group TL0) was also tested for comparison (control).

### 2.2. Fourier-Transform Infrared (FTIR) Spectroscopy

The absorbance spectra of the three materials (TL0, TL5, and TL10) were recorded using attenuated total reflectance-Fourier transform infrared (ATR-FTIR) spectroscopy (IRPrestige-21, Shimadzu, Kyoto, Japan) at a range of 4000–700 cm^−1^.

### 2.3. Contact Angles and Mechanical Properties

To measure the water contact angles (CAs) of the three materials, disc-shaped (8 mm in diameter, 1 mm in thickness) specimens were fabricated with a Teflon mold (*n* = 6/group). The pastes were filled into the molds and light-cured on both flat surfaces. The CAs of water (3 μL) on the cured material surfaces were recorded.

To test the flexural properties of the three materials, bar-shaped (10 × 2 × 1 mm) specimens were prepared using stainless steel molds (*n* = 6/group). A three-point bending was conducted with a universal testing machine (UTS, 3343, Instron, Canton, MA, USA) at 1.0 mm/min. The flexural strength (*σ*_F_) in MPa was calculated as
*σ*_F_ = (3*FL*)/(2*bd*^2^)(1)
where *F* is the fracture load (N), *L* is 20 mm, and *b* and *d* are the width and thickness of the specimen in mm, respectively [5]. In addition, the flexural modulus (*E*) in GPa was obtained as
*E* = [(*FL*^3^)/(*δ*4*bd*^3^)] × 10^−3^(2)
where *F*/*δ* is the slope of the curve (N/mm) [5].

To test the compressive properties, cylinder-shaped (4 mm in diameter, 6 mm in height) specimens were fabricated as described above (*n* = 6/group). A compressive test was done at 1.0 mm/min. Compressive strength (*σ*_C_) in MPa was calculated as
*σ*_C_ = *F*/*A*(3)
where *A* is the specimen’s cross-sectional area (mm^2^).

### 2.4. Shear Bond Strength Test

For SBS test, 72 non-carious human premolars were collected under a protocol (BMRI 74005-452) by the Institutional Review Board of Kyungpook National University Hospital (Daegu, Korea). Parallel-sided crown segments were prepared using a low-speed diamond saw, and the dentin discs were embedded using an acrylic resin. A smear layer was formed on the surface with a 600-grit wet silicon carbide paper. The dentin specimens were assigned into six groups depending on the type of the TheraCal LC materials (groups TL0, TL5, and TL10) and time point of debonding (immediate and 24 h after bonding).

The embedded dentin specimens were secured in a metallic device for the cement application and were mounted underneath a split Teflon mold with a central orifice (3.0 mm in diameter). Just before bonding, the surface had been blot-dried with a piece of lint-free paper to keep it moist. The TheraCal LC materials were inserted into the mold and light-cured.

For each TheraCal LC group, half of the bonded specimens (*n* = 12/group) were immediately tested using the UTS (immediate bond strength). A thin wire was looped around the cylinder base, and the shear load was applied at 1.0 mm/min until failure. The SBS values in MPa were calculated from the maximum load at failure divided by the bonding area. For each TheraCal LC group, the remaining half (*n* = 12/group) were tested after storage at 37 °C and 100% humidity for 24 h (24-h bond strength). All fractured surfaces were observed with optical microscopy to classify the failure modes.

### 2.5. Material–Dentin Interface Evaluation

One bonded specimen for each TheraCal LC material was additionally prepared and embedded using an epoxy resin. After curing of the resin, each embedded specimen was cross-sectioned using the low-speed diamond saw to expose the cement–dentin interface. The polished specimens were treated with 6 N HCl for 5 sec and then with 5% NaOCl for 5 min [6]. All the specimens were cleaned, dried, mounted on stubs, and gold-coated prior to scanning electron microscopy (SEM, JSM-6700F, Jeol, Tokyo, Japan).

### 2.6. Ion Release Analysis

Disc-shaped specimens (8.0 mm in diameter, 1.6 mm in thickness) were prepared using polyvinyl chloride molds (*n* = 6/group). The pastes were compacted into molds and then light-cured on both the top and bottom surfaces. The discs were placed on the bottom of glass scintillation vials with 10 mL of water, sealed, and stored at 37 °C, and the water was collected and renewed after 3 h and 1, 3, 7, 14, 21, and 28 days [1]. The pH and calcium ion of the medium were monitored using a pH probe (InLab Routine Pro, Mettler-Toledo, Schwerzenbach, Switzerland) and a calcium ion selective electrode (perfectION, Mettler-Toledo) connected to a pH/Ion meter (SevenCompact, Mettler-Toledo), respectively. To measure the released calcium ion, the solution was supplemented with ionic strength adjuster after the pH measurement.

### 2.7. Statistical Analysis

The results of the CA, mechanical properties, and SBS were analyzed using a two-way (two variables: materials and debonding time) and one-way analysis of variance (ANOVA), respectively. Post hoc comparisons were done using the Tukey test (*α* = 0.05). Since neither pH nor Ca release data met the assumptions of normality and homogeneity of variance, the difference between the value at each time and the value at 28 days (pH) or 7 days (calcium) within each material was analyzed with the Wilcoxon signed rank test. The pH/Ca data at each specified time among the three materials was analyzed with the Mann–Whitney test. The significance levels were adjusted using the Sidak correction.

## 3. Results

### 3.1. FTIR Spectra

Figure 1 presents the FTIR spectra of di-HEMA-phosphate and three TheraCal LC materials. The peak heights assigned to P-O-R stretch increased as di-HEMA-phosphate was added into the original material.

### 3.2. Contact Angles and Mechanical Properties

The CAs and mechanical properties of the unmodified and modified materials are shown in Table 1. The water CAs were significantly decreased by increasing the volume of di-HEMA-phosphate added into the original material (*p* < 0.001). The addition of the acidic monomer into the original material did not significantly alter the flexural strength nor compressive strength values (*p* > 0.05). On the contrary, higher concentrations of incorporated monomer significantly increased the flexural moduli (*p* < 0.001), indicating that its addition made the material stiffer.

### 3.3. Shear Bond Strengths and Failure Modes

The SBS values and corresponding failure modes of the three materials at the two different debonding times are shown in Table 2. The two-way ANOVA revealed significant differences among materials (*p* < 0.001). However, the debonding time (*p* = 0.600) and interaction between the two variables (*p* = 0.677) were not significantly different. The Tukey post hoc analysis revealed that the addition of 5.0 wt% di-HEMA-phosphate into the material (TL5) significantly increased the SBS values (*p* < 0.001) to dentin, as did the 10 wt% addition (TL10; *p* < 0.001). The control group showed exclusively adhesive failures, while the occurrence of mixed failures increased in the TL5 and TL10 groups, the failure mode being the most predominant in the TL10 group.

### 3.4. Material–Dentin Interfaces

Figure 2 presents the SEM images of the material–dentin interfaces for the three groups. The interface of the control group indicates poor adaptation and wetting of the original material to the dentin surface. The incorporation of the adhesive monomer di-HEMA-phosphate into the material (5 or 10 wt%) resulted in more intimate contact and hybrid layer formation along the interfaces, together with the formation of tag-like structures. This aspect was more prominent in the TL10 group than the TL5 group.

### 3.5. pH and Calcium Ion Release

Table 3 summarizes the pH values and released calcium ions. All three materials showed a rapidly elevated pH (alkaline) at 3 h of immersion (*p* < 0.05). Thereafter, the materials showed no further significant increases in pH after 3 h when compared to the values at 28 days (*p* > 0.05). Among the three materials, there were no significant differences in pH at each immersion time (*p* > 0.05). For all three materials, the leaching of calcium ions from the soaking water gradually increased to a maximum at 7 days and then decreased over 28 days. The calcium releases at 3 and 14 days were not significantly different from those at 7 days for the three materials (*p* > 0.05). From 3 days, the TL10 group released a significantly greater amount of calcium ions than did the other two groups (*p* < 0.05). In the TL10 group, the highest calcium release was maintained until the final experimental period (*p* < 0.05).

## 4. Discussion

In clinical practice, TheraCal LC is directly applied to dentin without the use of any adhesive. Thus, its initial bond and stability to dentin is very important for guaranteeing firm subsequent restorative procedures. In this study, the SBS values of the original material were nearly zero MPa (Table 2), although the material was applied to moist dentin following the manufacturer’s recommendations. The use of adhesives prior to TheraCal LC application can definitely improve dentin bond strengths, as reported by Wang and Suh [3]. However, in the in vitro study, calcium and hydroxyl ion releases from the material coated with adhesives were adversely diminished.

Therefore, in this study, di-HEMA-phosphate, which is commonly used in commercial self-etch adhesives, was incorporated into the TheraCal LC at two concentrations (Figure 1), and the effect of its addition on dentin bond strengths was evaluated. The acidic functional monomer has a hydrogen phosphate that is capable of decalcifying and chemically adhering to hydroxyapatite simultaneously [7]. It was found that the materials containing the functional monomer at 5.0 and 10.0 wt% had significantly higher SBS values than did the original version (Table 2). Therefore, the first null hypothesis, that the functional monomer addition into the material would not enhance the SBS values, was rejected. The mechanical property and ion release results of the unmodified and modified materials showed that incorporation significantly increased the flexural moduli (Table 1) and calcium ion release (Table 3), but it did not affect the other properties. Therefore, the second null hypothesis, that the functional monomer addition would not affect the mechanical properties nor the ion release tendency of the material, was partially rejected.

Most bond strength tests are performed 24 h after the bonding procedure, specimens being stored in water (or 100% humidity). However, in most clinical situations, a command set of the light-cured pulp-capping material facilitates immediate subsequent restoration procedures, leaving the material subjected to various stress immediately after bonding. Initial (immediate) bond strengths should be high enough to resist various stresses that will be applied to the pulp-capping material during the placement of a restoration material. Therefore, in this study, an immediate SBS test was performed, together with a 24-h SBS test (Table 2). The results show that the modified materials attained higher SBS values and more favorable failure patterns than the original one at both debonding times. In addition, better adhesion was achieved by increasing the functional monomer content. It can be speculated that the modified versions form a relatively strong adhesion to the dentin surface immediately after light curing.

As seen in Figure 2, the materials containing the functional monomer (TL5 and TL10) showed greatly improved material–dentin interfacial interaction, without significant gap formation at the interface [4]. This finding may be attributed to their enhanced wetting and spreading to the smear layer-covered dentin surface and their penetration due to forming tag-like structures that would anchor the material micromechanically to the underlying dentin. These findings also suggest that the adhesive TheraCal LC could be used as a secure pulp-capping material in terms of bonding for subsequent restorative steps.

The incorporation of the functional monomer into TheraCal LC might jeopardize the mechanical properties of the original material, thus adversely affecting its fracture resistance. In this study, the flexural and compressive strengths were not significantly affected by the addition of the functional monomer (Table 1). Only the flexural moduli, the materials’ resistance to elastic deformation during bending, increased with greater concentrations. It seems that the di-HEMA-phosphate monomer, which has reactive double bonds at each end of the molecule, partly participated in cross-linking during polymerization, thus resulting in a slight increase in the value. The increased flexural moduli for the modified materials may provide enhanced resistance to the elastic deformation of the materials during placement and condensation of an overlying restorative material.

In this study, the adhesive monomer di-HEMA-phosphate was incorporated into TheraCal LC primarily to enhance its bonding ability. However, such compositional change may also alter its chemical properties. The present study re-confirmed the alkalinizing activity and calcium-releasing ability of TheraCal LC, which are caused by the hydration of the calcium silicate material [1,2]. Such calcium and hydroxyl ion releases are beneficial for inducing pulp tissue repair and calcific barrier formation [8]. Although all the test materials released calcium ions and maintained an alkaline pH over 28 days (Table 3), the TL10 material released significantly higher calcium ions than the other materials after 3 days and up to 28 days. The hydration of TheraCal LC may be compromised because the material itself does not include water. However, it seems that a polymerizable solubility enhancer, polyethelyene glycol dimethacrylate (PEGDMA), in composition and post-polymerization water sorption allow it to uptake water from the environment and diffuse water within the material, consequently promoting the material’s hydration [1]. Moreover, the addition of di-HEMA-phosphate made the material more hydrophilic (Table 1), which promoted ion release through the enhanced inward/outward movement of water [8]. Increased free water uptake from the moist dentin into the more hydrophilic resin matrix may have eventually promoted the formation and outflow of portlandite (calcium hydroxide), resulting in greater calcium release [8]. Di-HEMA-phosphate was effective in enhancing calcium release only when incorporated in relatively high concentration (10.0 wt%).

The results of this in vitro study suggest that the incorporation of an acidic monomer (di-HEMA-phosphate) into TheraCal LC improves its adhesion to dentin surface without negatively compromising the mechanical and chemical properties. The 10.0 wt% addition resulted in rather greater calcium ion release than in the control over the experimental period. However, other properties of the acidic monomer-modified TheraCal LC, including the degree of conversion, water sorption/solubility, and cytotoxicity, were not tested in this study. In addition, only one acidic monomer was tested. TheraCal LC has been reported to present unfavorable pulpal responses; this was probably due to the unpolymerized resin monomers in the material, which may exert their toxic effects [9]. Although the addition of the acidic monomer did not negatively affect the mechanical properties (Table 1), the cytocompatibility of the modified materials definitely requires further investigation. Finally, in vitro and in vivo bioactivity tests should be conducted to confirm the clinical efficacy of self-adhesive TheraCal LC calcium silicate material.

## Figures and Tables

**Figure 1 materials-13-00293-f001:**
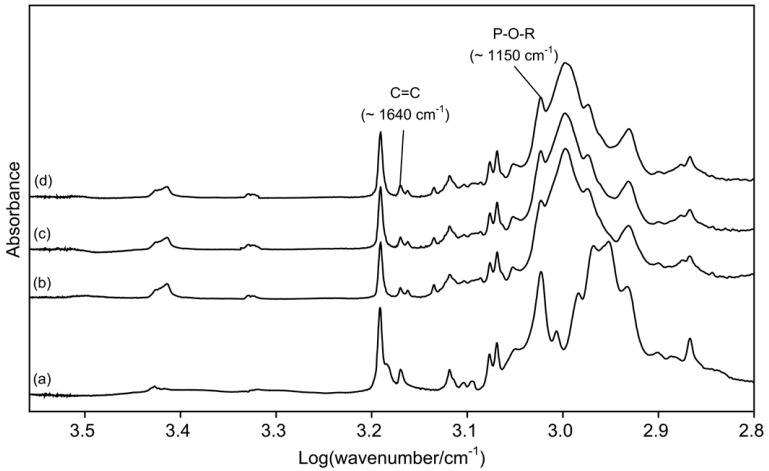
Fourier transform infrared (FTIR) spectra of di-2-hydroxyethyl methacryl hydrogenphosphate (di-HEMA-phosphate) (a) and three TheraCal LC materials (b), TL0; (c), TL5; and (d), TL10).

**Figure 2 materials-13-00293-f002:**
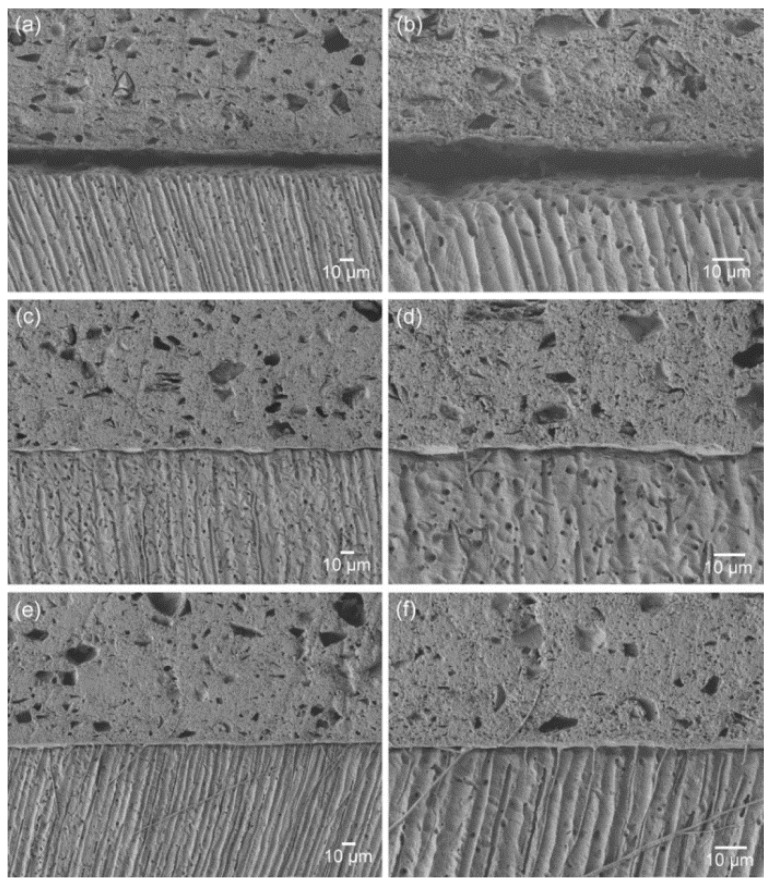
SEM images of the material-dentin interfaces: (**a**), (**b**) TL0; (**c**), (**d**) TL5; and (**e**), (**f**) TL10 (**a**), (**c**) and (**e**), ×450; (**b**), (**d**) and (**f**): ×1000).

**Table 1 materials-13-00293-t001:** Contact angles and mechanical properties of three TheraCal LC materials (means (standard deviations), *n* = 6/group).

Group Code	Contact Angle (°)	Flexural Properties		Compressive Strength (MPa)
		Flexural Strength (MPa)	Flexural Modulus (GPa)	
TL0	62.3 (2.4) ^a^	34.4 (2.7) ^a^	1.8 (0.2) ^a^	145.6 (12.5) ^a^
TL5	41.0 (3.2) ^b^	31.3 (3.7) ^a^	2.5 (0.3) ^b^	139.7 (13.3) ^a^
TL10	25.9 (1.9) ^c^	35.6 (4.8) ^a^	3.4 (0.3) ^c^	145.1 (17.6) ^a^

Within a column, different superscript lower case letters (a, b, and c) indicate statistical difference (*p* < 0.05).

**Table 2 materials-13-00293-t002:** Shear bond strengths and failure modes of three TheraCal LC materials at two different debonding times (means (standard deviations), *n* = 12/group).

Group Code	Immediate Debonding				Debonding after 24 h				
	Bond Strength (MPa)	Failure Mode			Bond Strength (MPa)	Failure Mode			
		A	C	M		A	C	M	
TL0	0.12 (0.25)	12	0	0	0.09 (0.20)	12	0	0	c
TL5	2.34 (0.74)	9	0	3	2.60 (0.61)	7	0	5	b
TL10	5.18 (0.89)	2	0	10	5.31 (0.85)	2	0	10	a
	A				A				

For the bond strength values, different letters (capital (A), among columns; lower case (a, b, and c), among rows) indicate statistical difference (*p* < 0.05). A, adhesive failure; C, cohesive failure; M, mixed failure.

**Table 3 materials-13-00293-t003:** pH of the soaking water and calcium ion release (ppm) in the soaking water (means (standard deviations), *n* = 6/group).

Group Code	0 h	3 h	1 Day	3 Days	7 Days	14 Days	21 Days	28 Days
pH								
TL0	7.2 (0.1) ^a^	10.5 (0.4) ^a^	11.0 (0.5) ^a^	11.1 (0.5) ^a^	11.0 (0.6) ^a^	11.0 (0.3) ^a^	10.8 (0.3) ^a^	10.7 (0.5) ^a^
TL5	7.2 (0.2) ^a^	10.4 (0.2) ^a^	10.7 (0.5) ^a^	10.8 (0.5) ^a^	11.0 (0.3) ^a^	10.9 (0.6) ^a^	10.9 (0.3) ^a^	10.6 (0.6) ^a^
TL10	7.2 (0.2) ^a^	10.3 (0.5) ^a^	10.8 (0.4) ^a^	11.0 (0.6) ^a^	11.3 (0.5) ^a^	11.1 (0.4) ^a^	11.1 (0.7) ^a^	11.0 (0.4) ^a^
Calcium								
TL0	0.2 (0.1) ^a^	1.9 (0.4) ^a^	3.4 (0.4) ^a^	3.9 (0.6) ^a^	4.3 (0.6) ^a^	4.1 (0.5) ^a^	3.2 (0.4) ^a^	1.8 (0.6) ^a^
TL5	0.2 (0.1) ^a^	1.5 (0.3) ^a^	3.1 (0.4) ^a^	4.5 (0.6) ^a^	5.1 (0.6) ^a^	5.1 (0.8) ^a^	3.5 (0.7) ^a^	2.7 (0.6) ^a^
TL10	0.2 (0.1) ^a^	1.6 (0.4) ^a^	3.6 (0.5) ^a^	6.4 (0.8) ^b^	7.5 (0.9) ^b^	7.4 (0.8) ^b^	5.2 (0.6) ^b^	4.7 (1.0) ^b^

Within a row, each underlined value is significantly different from the value at 28 days (pH) or 7 days (calcium) (*p* < 0.05). Within a column, different superscript lower case letters (a and b) indicate statistical difference (*p* < 0.05).

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
