# Peer review of "Dentin Bonding of TheraCal LC Calcium Silicate Containing an Acidic Monomer: An In Vitro Study"

_materials, 2020, doi:10.3390/ma13020293_

Round 1

Reviewer 1 Report

Dear Editor,

Regarding the submitted manuscript “Improved Dentin Bonding of TheraCal LC Calcium Silicate by Incorporating an Acidic Monomer” this review will be divided in overall and detailed appreciation.

The presented study is intended to be an in vitro study to evaluate TheraCal Properties after adding an Acidic Monomer.

The article is well written, and the proposed objectives are attained. It is this referee belief that only minor changes are needed before considering the manuscript for publication:

Title should be revised since it should not present the results and it should mention that it is an in vitro study. Something like: Dentin Bonding of TheraCal LC Calcium Silicate by Incorporating an Acidic Monomer – An in vitro study "On the contrary, higher concentrations 160 of incorporated monomer increased the flexural moduli more and more (p < 0.001), indicating that its 161 addition made the material a little stiffer." – Please reformulate in a more scientific way. Table 3 – The results are presented as? Mean +/- SD? Discussion – The authors should address in more detail the cytotoxicity possibility of adding more monomers into the Theracal

Author Response

The article is well written, and the proposed objectives are attained. It is this referee belief that only minor changes are needed before considering the manuscript for publication:

- We really appreciate your encouraging comment.

Title should be revised since it should not present the results and it should mention that it is an in vitro study. Something like: Dentin Bonding of TheraCal LC Calcium Silicate by Incorporating an Acidic Monomer – An in vitro study.

- We agree with your suggestion. The title has been modified according to your comment.

"On the contrary, higher concentrations of incorporated monomer increased the flexural moduli more and more (< 0.001), indicating that its addition made the material a little stiffer." – Please reformulate in a more scientific way.

- Thank you for your comment. We have removed subjective terms from the sentence.

Table 3 – The results are presented as? Mean +/- SD?

- We appreciate you pointing this out and have modified the table caption.

Discussion – The authors should address in more detail the cytotoxicity possibility of adding more monomers into the Theracal. 

- Thank you for your suggestion. We have added some description about the cytotoxicity issue in the last paragraph of the Discussion section.

Reviewer 2 Report

The manuscript is generally well written. However, I have some suggestions.

The aim is not the description of what was done; it should be a short statement of the main point of the research (the reason why you are doing the research).

Material and method

How many samples were prepared for each test for each study and control group?

Please explain why you decided to use this number of samples for each test.

Lines 75-85

Please write each test in separate paragraph.

Please write equation in word format.

Conclusions

Conclusions are missing

Author Response

The manuscript is generally well written. However, I have some suggestions.

- Thank you very much for your encouraging comment and valuable suggestions.

The aim is not the description of what was done; it should be a short statement of the main point of the research (the reason why you are doing the research).

- We agree with you. According to the author instructions, the Abstract section should be a total of about 200 words maximum. Thus, we have added a short clause in the first sentence of the Abstract to clarify the aim of our study.

Material and method

How many samples were prepared for each test for each study and control group? Please explain why you decided to use this number of samples for each test.

- In this study, the sample size in the contact angle measurements, mechanical tests, and ion release analysis was 6. In the case of the shear bond strength test, it was increased to 12 because the values were rather low and, thus, we tried to decrease the risk of committing a type II error. In this revision, the sample sizes are more clearly addressed in the text and table captions.

Lines 75-85

Please write each test in separate paragraph. Please write equation in word format.

- The subsection has been modified according to your instructions.

Conclusions

Conclusions are missing

- According to the author guidelines, the Conclusions section is not mandatory. It seems that the last paragraph of the Discussion section (draft) clearly showed our conclusions. Thus, we did not prepare the separate Conclusions section. Once again, thank you very much for your valuable comments and suggestions.